The experiences of COVID-19 preprint authors: a survey of researchers about publishing and receiving feedback on their work during the pandemic

Rzayeva Narmin n.rzayeva@cwts.leidenuniv.nl 1 2 3
Henriques Susana Oliveira 1 2 4
Pinfield Stephen 1 5
Waltman Ludo 1 2
1 Research on Research Institute (RoRI) , London , United Kingdom
2 Centre for Science and Technology Studies (CWTS), Leiden University , Leiden , The Netherlands
3 Information Technologies and Systems Department, Azerbaijan University of Architecture and Construction , Baku , Azerbaijan
4 Central Library, Lisbon University Medical School , Lisbon , Portugal
5 Information School, University of Sheffield , Sheffield , United Kingdom
Stern David
Electronic publication date: 2023 Aug 22
Publication date: 2023
Volume: 11
Electronic Location ID: e15864
Received 2022 Dec 15; Accepted 2023 Jul 17
Copyright: ©2023 Rzayeva et al.
Copyright year: 2023
Copyright holder: Rzayeva et al.
License: This is an open access article distributed under the terms of the Creative Commons Attribution License, which permits unrestricted use, distribution, reproduction and adaptation in any medium and for any purpose provided that it is properly attributed. For attribution, the original author(s), title, publication source (PeerJ) and either DOI or URL of the article must be cited.
License URL: https://creativecommons.org/licenses/by/4.0/

Keywords: Preprints, Peer review, Covid-19 crisis, Scholarly communication, Survey, Feedback

Funding: Research on Research Institute Wellcome 221297/Z/20/Z Stephen Pinfield and Ludo Waltman were supported by Wellcome [221297/Z/20/Z] as part of its core funding of the Research on Research Institute (RoRI). The funders had no role in study design, data collection and analysis, decision to publish, or preparation of the manuscript.

==============================
The COVID-19 pandemic caused a rise in preprinting, triggered by the need for open and rapid dissemination of research outputs. We surveyed authors of COVID-19 preprints to learn about their experiences with preprinting their work and also with publishing their work in a peer-reviewed journal. Our research had the following objectives: 1. to learn about authors’ experiences with preprinting, their motivations, and future intentions; 2. to consider preprints in terms of their effectiveness in enabling authors to receive feedback on their work; 3. to compare the impact of feedback on preprints with the impact of comments of editors and reviewers on papers submitted to journals. In our survey, 78% of the new adopters of preprinting reported the intention to also preprint their future work. The boost in preprinting may therefore have a structural effect that will last after the pandemic, although future developments will also depend on other factors, including the broader growth in the adoption of open science practices. A total of 53% of the respondents reported that they had received feedback on their preprints. However, more than half of the feedback was received through “closed” channels–privately to the authors. This means that preprinting was a useful way to receive feedback on research, but the value of feedback could be increased further by facilitating and promoting “open” channels for preprint feedback. Almost a quarter of the feedback received by respondents consisted of detailed comments, showing the potential of preprint feedback to provide valuable comments on research. Respondents also reported that, compared to preprint feedback, journal peer review was more likely to lead to major changes to their work, suggesting that journal peer review provides significant added value compared to feedback received on preprints.

Introduction

The COVID-19 pandemic has arguably illustrated the value of open and rapid dissemination of scientific research on a global scale. All participants in the scholarly communication system, including research funders, publishers, infrastructure providers and research institutions, were prompted to reconsider their policies to respond to the public crisis, with most adopting more open approaches, aiming to make COVID-19 research openly or freely accessible. Early in the pandemic, a statement on “Sharing research data and findings relevant to the novel coronavirus (COVID-19) outbreak” (“Coronavirus [COVID-19]: sharing research data”) was issued by Wellcome (2020) and signed by 160 organisations worldwide. The statement called on actors in the research system to implement a set of agreed principles aimed at achieving greater openness. The statement included the following principle, designed to accelerate the dissemination of COVID-19 research and bypass the often-lengthy process of peer review carried out by journals:

“COVID-19 papers should be made available via open access preprint servers prior to publication in journals.”

In April 2020, a group of publishers and related organisations came together and launched the COVID-19 Rapid Review Initiative “to maximise the efficiency of peer review, ensuring that key work related to COVID-19 is reviewed and published as quickly and openly as possible” (Hurst & Greaves, 2021). The Research on Research Institute (RoRI), with which we are affiliated, was also part of the group. In December 2021, the group published an evaluation of its work, examining the extent to which the key commitments made at the beginning of the pandemic in the Wellcome statement had been realised (Waltman et al., 2021).

As part of this evaluation, we conducted a survey study aimed at developing a better understanding of the experience of COVID-19 preprint authors with respect to the following:

1. Preprint posting. When did the authors of COVID-19 preprints publish their first preprints? What were their primary motivations for preprinting their COVID-19 work? And what were their intentions regarding preprinting future research outputs?

2. Feedback on preprints. What proportion of COVID-19 preprints received feedback? What were the main channels of feedback? And what was the nature and impact of the feedback?

3. Journal peer review. To what extent and in what ways did comments made by editors and reviewers of journals change papers? How did the effect of journal peer review differ from the effect of feedback on preprints?

Related work

A number of studies show that increasing attention is being given to preprints, with new servers being set up and preprints being seen as enabling early and rapid dissemination of research outputs (Abdill & Blekhman, 2019; Chiarelli et al., 2019; Chung, 2020; Delfanti, 2016; Polka, 2017; Puebla, Polka & Rieger, 2022; Smart, 2022; Vale, 2015). The increasing adoption of preprinting has raised questions regarding the motivations of authors who choose to engage in preprinting. Multiple studies have surveyed or interviewed preprint authors, research funding representatives, research-conducting organisations, and other stakeholders (ASAPbio, 2020; Chiarelli et al., 2019; Fraser, Mayr & Peters, 2021). These studies consistently identified the rapid dissemination of research outputs as the primary motivation for preprint adoption among participants. Additionally, Fraser, Mayr & Peters (2021) reported that preprint authors increasingly cited the enhanced visibility of their research as a motivation for preprint posting. Similarly, ASAPbio (2020) found that the majority of their respondents were motivated by the free accessibility of preprints.

It has been found that authors also place great value on the opportunity to receive feedback on their work when posting it as a preprint (Chiarelli et al., 2019; Ginsparg, 2016). However, previous studies found relatively low levels of feedback on preprints (Carneiro et al., 2022; Kodvanj et al., 2022; Malički et al., 2021). According to Sawyer et al. (2022), critiques and questions for authors were the most frequently occurring type of comments. Carneiro et al. (2022) and Malički et al. (2021) also examined comments on preprints and evaluated their impact. They reported that the majority of the comments consisted of criticisms, corrections, suggestions, or compliments about the research, leading them to conclude that, despite the low rate of commenting, comments still “address content that is similar to that analyzed by traditional peer review” (Carneiro et al., 2022) and “may have potential benefits for both the public and the scholarly community” (Malički et al., 2021). However, the authors also noted that additional research is necessary to assess the influence of these comments on scientific works and on the journal publication process.

The COVID-19 pandemic created a pressing need to rapidly share research outputs globally (Horbach, 2020; Horbach, 2021). As a means of ensuring swift open access to and feedback on research, preprints were embraced by many authors of COVID-19 research, often using preprint servers as a complement to peer-reviewed journals (All that’s fit to preprint, 2020; Hook & Porter, 2020; Callaway, 2020; Taraborelli, 2020; Fraser et al., 2021). In earlier work (Waltman et al., 2021), we showed that almost 40,000 COVID-19 preprints were posted between January 2020 and April 2021. In the early months of the pandemic, the number of COVID-19 preprints even exceeded the number of COVID-19 publications in peer-reviewed journals.

The widespread adoption of preprinting during the pandemic prompted discussion regarding the credibility of research posted on preprint servers–research that typically has not been peer reviewed (Carneiro et al., 2020; Nabavi Nouri et al., 2021; Smart, 2022; Soderberg, Errington & Nosek, 2020). Addressing this issue, several recent studies compared preprints with the corresponding articles published in peer-reviewed journals. Brierley et al. (2022) compared abstracts of preprints and abstracts of the corresponding journal articles, observing that many journal article abstracts were not significantly different from their preprint counterparts. Zeraatkar et al. (2022) reported a comparison of the key methods and results in preprints and the corresponding journal articles, finding “no compelling evidence that preprints provide less trustworthy results than published papers”. Janda et al. (2022) found that the majority of clinical studies that were initially posted as preprints on medRxiv and later published in peer-reviewed journals demonstrated consistent and robust study characteristics, findings, and overall conclusions.

Methods

Survey overview

To gather opinions on these issues, we surveyed authors of COVID-19 preprints posted on arXiv, bioRxiv, medRxiv and ChemRxiv in 2020. The survey was carried out using the Qualtrics XM software. We presented the authors with 36 questions about their experience of posting their paper on a preprint server and submitting it to a journal. However, the total number of questions answered by any individual participant varied depending on which ‘pathway’ they took through the survey based on several factors, such as whether or not their paper had been published in a journal. The questions in the survey were grouped into three main sections:

• Demographic questions–including the country in which the authors’ were based, type of research institution, research experience, and gender.

• Experience of preprinting–including motivations and future intentions concerning preprinting, channels through which feedback on the preprint was received, and changes in preprints resulting from the feedback.

• Experience of journal submission and the journal peer review process–including the opinion of the authors about the differences in peer review of COVID-19 papers compared with their former experience of article publishing and changes made to the paper in response to journal peer review.

The survey included both mandatory and optional questions of different types. Some were simple closed questions, such as “Is your paper now published in a peer-reviewed journal?”, to which respondents could respond “Yes” or “No”. There were also multiple choice and Likert scale questions. These included a “Don’t know/can’t remember/not applicable” or “Other, please specify in the box below” option. Respondents were also able to revise their responses by clicking the “Back” button in the online survey tool.

The survey was piloted by ten people: two individuals affiliated with journal publishers, two affiliated with preprint service providers, and six researchers recruited by the project team. A copy of the questionnaire and survey data is available in figshare (Rzayeva et al., 2023). We obtained ethical approval to conduct the survey from the Ethics Review Committee of the Faculty of Social and Behavioural Sciences at Leiden University. Participants were informed about this on the survey’s title page. Prior to taking the survey, respondents were asked to provide their consent by clicking the corresponding button. Additionally, the title page informed respondents that we would not collect any personal data about them.

Data collection and analysis

The titles of COVID-19 preprints posted in 2020 and the contact details of their authors were collected directly from the four preprint servers, by manually copying the information from the page of each preprint. For bioRxiv and medRxiv we considered all preprints in the collection “https://connect.biorxiv.org/relate/content/181”. In the case of arXiv, we considered all preprints in the collection “COVID-19 SARS-CoV-2 preprints from arXiv”. Because ChemRxiv did not provide such a COVID-19 collection, we created a search query and identified all preprints posted in 2020 whose title or abstract included “coronavirus”, “covid-19”, “sars-cov”, “ncov-2019”, “2019-ncov”, “hcov-19”, or “sars-2”.

Some researchers were corresponding author of multiple COVID-19 preprints. We therefore deduplicated the names of corresponding authors using the “Remove Duplicates” option in Microsoft Excel. In case of multiple preprints with the same corresponding author, only the first preprint was retained. This yielded a list of 12,230 COVID-19 preprint titles, names of the corresponding authors, and their email addresses. 12% of the preprints were from arXiv, 17% from bioRxiv, 62% from medRxiv, and 9% from ChemRxiv. We then emailed the corresponding authors, inviting them to participate in our survey. They were sent a personal link to take part in the survey. Personal links provided by Qualtrics XM prevented participants from completing the survey multiple times. We did not offer any incentives to participate in the survey, and we did not contact potential participants other than by email. For instance, we did not advertise the survey on the Internet. Invitations were sent in batches between May 20 and May 26, 2021. Of the 12,230 invitations, 305 bounced and 523 failed, so 11,402 invitations were sent successfully. The survey stayed open for 55 days. It was closed on July 14, 2021. The survey data was stored in Leiden University’s cloud storage.

Of the 11,402 corresponding authors that received an invitation to participate in the survey, 826 chose to participate, resulting in a participation rate of 7%. 673 participants completed the survey, giving a completion rate of 81%. Of the 673 respondents, 18.4% had posted their preprint on arXiv, 14.1% on bioRxiv, 57.7% on medRxiv and 6.8% on ChemRxiv. A total of 3.0% of the respondents answered that they had posted their preprint on a different platform. These respondents probably misunderstood the question, since their preprint had in fact been posted on one of the four preprint servers considered in our study. Alternatively, it could be that some respondents had posted their preprint on multiple servers.

To calculate confidence intervals for the proportions or percentages reported in this article, we used the normal approximation of the binomial proportion confidence interval. Confidence intervals are reported at a 95% confidence level and denoted by the ± sign. In addition to quantitative data, the survey yielded qualitative responses through open-ended questions and comment boxes. These responses were coded using ATLAS.ti. Extracts from qualitative responses quoted below have been lightly edited, for instance to correct typographical errors.

Characteristics of respondents

Of the survey’s respondents, 516 (76.7%) described their gender as “man”, 137 (20.4%) as “woman”, 16 (2.4%) responded “prefer not to say”, and four (0.6%) responded “prefer to self-describe” (free-text responses included “non-binary” and “genderfluid”). Responses were received from researchers with different levels of experience in conducting research (Fig. 1); 126 (18.7%) had up to 5 years of experience, 381 (56.6%) had 6–25 years of experience and 162 (24.1%) had 25 years or more experience (with four respondents answering “not applicable”). Responses were received from a total of 78 countries. The country with the most respondents was the USA, with 131 (19.5%) responses, followed by 93 (13.8%) responses from the UK, 70 (10.4%) from India, and 33 (4.9%) from Brazil. A total of 283 (42.1%) responses were received from Europe, 169 (25.1%) from North America, 131 (19.5%) from Asia, 51 (7.6%) from South America, 21 (3.1%) from Africa, and 15 (2.2%) from Australasia (with three not disclosing their country). Most respondents, 424 (63.0%), were based in universities or colleges, with a further 91 (13.5%) in hospitals or medical schools. Smaller numbers were based in other organisation types, comprising public research organisations (56, 8.3%), governments (28, 4.2%), industrial or commercial organisations (27, 4.0%), non-governmental organisations (17, 2.5%), and “other” kinds of organisations (30, 4.5%).

Figure 1 Characteristics of respondents (n = 673).

Results

Preprint posting

Notably, the vast majority of the survey participants first engaged in preprinting during the pandemic: of the n = 673 participants, 66.7% first posted a preprint during 2020 or 2021, 14.9% between 2017 and 2019, and 18.4% before 2017. medRxiv, the first preprint server specialising in health sciences, was launched in 2019, just before the start of the pandemic. This is likely to be one of the reasons why many respondents had limited experience with preprinting.

For each of the four preprint servers considered in our survey, Fig. 2 shows the percentage of survey participants who had not posted any preprints before 2020. Although only authors whose preprints were posted on the four preprint servers participated in our survey, some (n = 18) indicated other platforms where their preprint was located (such as SSRN, Research Square, etc). We do not know the reason for this, but it is possible that their preprint was deposited on more than one server.

Figure 2 The percentage of survey participants who had not posted any preprints before 2020, broken down by preprint server (n = 673).

To understand authors’ motivations for preprinting their COVID-19 work, the survey listed possible motivations based on the findings of a previous study (Chiarelli et al., 2019). Authors were invited to choose one or more features that motivated them to preprint their COVID-19 work. The survey results showed that authors (n = 673) were primarily motivated to “achieve early and rapid dissemination” of their work (86.2% ± 2.6%), make their work “openly available” (63.9% ± 3.6%) and “increase opportunities for feedback from readers” (31.2% ± 3.5%). Of all respondents, 97.0% chose at least one of these advantages of preprinting as their main motivation. Further details are provided in Fig. 3.

Figure 3 Motivations of survey respondents for preprinting their COVID-19 research (n = 673).

Additional comments regarding the importance of achieving early and rapid dissemination during the pandemic were expressed by 25 survey participants, with different levels of experience of preprinting:

“We felt it was important to make our research available in a timely manner as the peer-review process can be time consuming.” (Author with 3–5 years of research experience who first preprinted their work in 2020/2021.)

“Wanted to get this work on vaccine design disseminated as early as possible for further work in urgent vaccine development.” (Author with more than 25 years of research experience who first preprinted their work in 2020/2021.)

Among all respondents (n = 673), 83.1% (± 2.8%) declared that they intended to post at least some of their work on a preprint server in the future. Just 6.4% (± 1.8%) of respondents declared that they did not intend to post future work on a preprint server. We were also specifically interested in the future intentions of new contributors to preprint servers. Among authors who first posted their work on a preprint server during the pandemic (n = 449), 78.0% (± 3.8%) also planned to preprint at least some of their future works. This may suggest that the pandemic, possibly in combination with other developments (e.g., growing adoption of open science practices), will lead to long-term changes in authors’ preprinting practices.

Feedback on preprints

Since receiving feedback is often argued to be one of the key advantages of posting preprints (Chiarelli et al., 2019; Ginsparg, 2016), we asked authors of COVID-19 preprints about the feedback they received on their work after posting it on a preprint server. The survey specified that “by feedback, we mean comments or questions received in response to the preprint as posted on a preprint server” (excluding “any formal responses from peer reviewers received as part of the process of submitting the paper to a journal”). Of the respondents (n = 673), 52.8% (± 3.8%) reported receiving feedback on their preprint. Kodvanj et al. (2022) found that only 18% and 12% of COVID-19 preprints posted on bioRxiv and medRxiv, respectively, had comments. They considered only comments posted directly on the preprint server. Ross et al. (2022) reported an even lower rate of feedback for medRxiv preprints posted within 36 months from 2019. They found that 8% of the preprints had received at least one comment. The differences between our findings and those of others seem to be related to the fact that unlike other studies, we directly surveyed authors, which enabled us to consider diverse types of public and private feedback on preprints. Other studies considered only specific forms of public feedback, in particular comments on preprint servers. The definition of feedback adopted in our survey therefore is broader than the definitions used in other studies. It includes a wide range of feedback channels, e.g., emails to the author, rather than just public comments on the preprint server. In contrast, Kodvanj et al. (2022) estimated the extent of public peer review using Disqus comments and Altmetric data as proxies. Ross et al. (2022) used data from the medRxiv website, internal medRxiv data, plus data from Altmetric.com.

We also analysed the breakdown by preprint server. As Fig. 4 illustrates, the percentage of preprints that received feedback was highest for bioRxiv, though the differences between platforms were minimal.

Figure 4 The percentage of COVID-19 preprints that received feedback, broken down by preprint server (n = 673).

Error bars represent 95% confidence intervals.

To learn more about the channels for receiving feedback, we asked authors how they received comments on their preprint. Results reported by the survey participants are illustrated in Fig. 5. In our analysis, we grouped the channels into two groups–“open” and “closed” ways of receiving feedback. We determined feedback to be “open” when it was publicly available, e.g., given directly on a preprint server 19.6% ± 3.2%), peer review platform (3.3% ± 1.5%), or social media (18.7% ±3.2%). Feedback was classified as “closed” when it was given privately to the author, e.g., by email (41.9 ±4.0%) or during meetings with colleagues (13% ± 2.7%). Overall, 54.9% (± 4.1%) of all feedback on preprints reported by respondents was given in a “closed” way, whereas 41.6% (± 4.0%) of feedback was given in an “open” way.

Figure 5 Main channels of feedback for authors of COVID-19 preprints (n = 355).

Error bars represent 95% confidence intervals.

Notably, bioRxiv authors (n = 56) reported that 50.0% (±13.1%) of feedback received on their preprint was given directly on the preprint server, while for medRxiv authors (n = 200), this was the case for 38.5% (± 6.7%) of feedback received. These results may suggest the value of the possibility to provide feedback on a preprint directly on the preprint server. arXiv and ChemRxiv do not provide this possibility. We were also interested in the nature of the feedback given, particularly whether it contained detailed comments. In the survey, we asked authors to report whether they received “detailed feedback on the research presented in the preprint”. Alternative options given in the survey were, “brief feedback (e.g., correcting a mistake)”, “comments with thanks for the paper, retweets, etc.”, “suggestions on extending the research in new areas for further research”, and “requests from other authors to cite their work”. In 25.4% ( ± 4.5%) of the cases, authors reported receiving detailed feedback on their research.

For each preprint server, we analysed the percentage of preprints for which the author considered the feedback received to be of a detailed nature. Figure 6 illustrates that bioRxiv (n = 56) had the highest percentage of detailed feedback, at 32.1% (± 12.2%). However, due to the limited number of responses per preprint server, statistical uncertainty is high, and we therefore cannot draw clear conclusions on differences between the various servers.

Figure 6 The percentage of COVID-19 preprints for which the feedback received was of a detailed nature, broken down by preprint server (n = 355).

Error bars represent 95% confidence intervals.

The final issue related to preprint feedback in the survey was the extent to which authors made changes to their preprints after receiving feedback. We asked authors to indicate to what extent changes had been made as a result of feedback on the different sections of their preprint. We listed the main sections and asked authors to indicate whether “minor changes” or “major changes” had been made. We did not attempt to define the terms “minor” and “major”, instead relying on participants’ perceptions and interpretations of the terms. Many participants were probably familiar with similar terminology that is commonly used in journal peer review processes. Figure 7 shows that major changes to preprints were uncommon. Only 7.8% (± 3.4%) (n = 243) and 5.3% (± 2.8%) (n = 244) of respondents who received feedback and answered this question declared that they had made major changes to the discussion/conclusion and results section of their preprint, respectively. For other sections, this percentage was even lower. As seen in Fig. 7, it was much more common for authors to make changes that they considered minor. The figure illustrates that most authors making changes to their preprint in response to feedback did not perceive the changes to be significant in altering their work.

Figure 7 Major and minor changes in COVID-19 preprints in response to feedback, broken down by different sections of the paper.

The number of respondents who evaluated each section differs, as participants were not required to assess every section. As a result, the sample size n for each section varies between 233 and 244.

Journal peer review

Submission to a journal

In our survey 87.2% of COVID-19 preprint authors (n = 673) had submitted their article to a peer-reviewed journal. The remaining 12.8% reported various reasons why their papers had not been submitted to a peer-reviewed journal. For instance, the authors (n = 86) reported that their article had not yet been completed (19.8% ± 8.4%), that they had not yet decided whether to publish their article in a peer-reviewed journal (18.6% ± 8.2%), and that their article required more feedback (14.0% ± 7.3%). However, the most common reason reported by authors for not submitting their article to a peer-reviewed journal is that they had decided not to do so (26.7% ± 9.4%). Some commented that the preprint was sufficient to draw attention to the study and noted shortcomings in the journal publication process, such as a long peer review time and expensive article publication charges:

“I do not intend to publish a paper; the intention was purely to attract the attention of public opinion and authorities for the importance of good practices like contact tracing to contain the pandemic.” (Author who first preprinted a work in 2020/2021, “the preprint was an individual action, the work is not connected to any institution/organization”.)

“The peer-review process is so long that the paper would no longer be timely by the time it was completed.” (Author who first preprinted a work before 2017, retired academic worker.)

“I have no possibility to pay for that. It was my own project.” (Author who first preprinted a work in 2020/2021, affiliated with university or college.)

Among authors who submitted their article to a peer-reviewed journal (n = 587), 61.8% (± 3.9%) reported that their paper had already been published by the journal. For authors whose articles had not yet been published in a peer-reviewed journal (n = 224), the most common reason, reported by 45.5% ( ± 6.5%), was that the article was still under review. The second most common reason, accounting for 30.8% (±6.0%), was rejection by the journal. The main reasons given for rejection were low quality and scope. At the same time, more than half of the authors who reported that their article was rejected stated that they intended to resubmit their article to another peer-reviewed journal.

Asking the authors (n = 587) about the order in which they had submitted their COVID-19 paper to a preprint server and a journal, we found that 53.0% (± 4.0%) first posted their work on a preprint server and then submitted it to a journal. A further 35.3% (± 3.9%) reported that posting the preprint and submitting the article happened at the same time. This aligns with an earlier survey conducted by Sever et al. (2019), covering authors of more than 4,000 bioRxiv preprints in 2019, which found that 42% of authors had posted their preprint before journal submission and 37% had posted concurrently with journal submission. This might be considered ‘conventional’ practices in relation to preprints, and is consistent with motivations reported by our respondents (e.g., early dissemination).

In the survey, we asked authors about the main features to which they paid attention in the choice of the journal to which they submitted their article. Figure 8 shows the proportion of respondents who selected “extremely important” or “very important” for each journal feature listed in the survey.

Figure 8 Journal features rated as “.extremely important” or “very important” by authors in the choice of the journal for their COVID-19 papers (n = 587).

The features considered in this question were based on earlier research that reported the results of an international survey of authors publishing in different types of journals, mega-journals and conventional journals (Wakeling et al., 2019). The findings from that study and the results of our survey are similar. Both groups of authors were primarily interested in publishing their works in what they saw as high-quality journals with a high-quality peer review process. The reputation of the publisher and the speed of peer review were also among the five most important journal features to the participants of both surveys. The comparison with Wakeling et al. (2019) illustrates that authors of preprints in our survey did not report noticeably different motivations in selecting journals.

Impact of peer review comments on papers

For preprints submitted to a journal, we further examined how the comments of reviewers invited by the journal prompted changes in the article. We asked authors whose COVID-19 articles went through a journal peer review process to what extent they had changed each section of their paper as a result of comments provided by reviewers and/or editors. Answers to this question are presented in Fig. 9.

Figure 9 The percentage of respondents who made major or minor changes to different sections of their COVID-19 article in response to comments provided by reviewers and/or editors.

The number of respondents who evaluated each section differs, as participants were not required to assess every section. As a result, the sample size n for each section varies between 352 and 359.

The authors reported major changes to approximately the same extent for the title, abstract and introduction sections of the article, accounting for around 10%of responses. For the methods section (n = 357), it was 13.2% (± 3.5%). A total of 19.1% (± 4.1%) of authors indicated making major changes to the results and 21.2% (± 4.2%) to the discussion/conclusion sections of their articles in response to comments of reviewers and/or editors. The percentage of minor changes reported by authors varied between 53% and 65% for most sections.

Our results with regard to abstracts are reasonably aligned with earlier research by Brierley et al. (2022), who compared abstracts between preprints and the corresponding journal articles for COVID-19 research in the first four months of the pandemic. According to the results of their study, 17% of journal article abstracts underwent major changes and over 50% minor changes compared with their preprint counterpart. In our survey (n = 355), this was the case for, respectively, 9.9% (± 3.1%) and 62.5% (± 5.0%) of the articles.

In the cases of both preprint feedback and journal peer review, the authors indicated that the results and discussion/conclusion sections were most subjected to major changes. We, therefore, compared the effect of preprint feedback and journal peer review on these two sections. Figure 10 shows that, according to the responses of surveyed authors, the journal peer review process resulted in a much higher rate of changes in articles than feedback on preprints.

Figure 10 The percentage of respondents who made major or minor changes to the results and discussion/conclusion sections of their COVID-19 article in response to preprint feedback and comments provided by reviewers and/or editors (n = 673).

To enable a balanced comparison, Fig. 10 reports percentages relative to the overall number of preprints included in our survey rather than percentages relative to the number of preprints that received preprint feedback (Fig. 7) or percentages relative to the number of preprints that received comments from reviewers and/or editors (Fig. 9).

We then asked the authors to what extent they had made specific types of changes to their papers to address comments made by reviewers and/or editors. The results are shown in Fig. 11. The authors reported that, in the case of journal peer review, major changes mainly consisted of further analysis, improving the presentation and readability of the article, and restructuring or reorganising the article.

Figure 11 The percentage of respondents who made specific types of changes (major or minor) in their COVID-19 article in response to comments of reviewers and/or editors.

The number of respondents who evaluated each item differs, as participants were not obliged to assess every item. As a result, the sample size n for each item varies between 357 and 359, except for the “Other” item, where n = 80.

To examine possible effects of initiatives of publishers to accelerate the publication process for COVID-19 papers (Horbach, 2020; Horbach, 2021), we also analysed whether authors experienced any differences in the speed, quality, and constructiveness of the journal peer review process for COVID-19 research compared with their prior experience in scientific publishing. Figures 12 and 13 show that, on average, authors did not experience any major differences.

Figure 12 Article processing time of COVID-19 papers compared to the former experience of survey respondents (n = 363).

Figure 13 Constructiveness and quality of journal peer review of COVID-19 papers compared to the former experience of survey respondents (n = 388 and n = 363).

For a better understanding, we divided the article processing time into three stages: the time of peer review, the time between submission of the revised version and acceptance, and the time to publication following acceptance. Of the respondents (n = 363), 35.5% (± 4.9%) reported that the time spent on peer review of their COVID-19 paper was shorter compared with their prior experiences, while 39.4% (± 5.0%) of the participants evaluated the peer review duration as longer. 39.7% (± 5.0%) agreed with the statement that “the time between submission of the revised version and acceptance was shorter than normal” for their COVID-19 paper, while 29.2% ( ± 4.7%) disagreed. The process of publishing an article after it was accepted was assessed as shorter than prior experiences by 46.8% (± 5.1%) of respondents, while 22% (± 4.3%) assessed it as longer (Fig. 12).

As for the quality and constructiveness of journal peer review, Fig. 13 shows that, on average, our respondents experienced peer review of their COVID-19 research to be slightly more constructive than peer review of earlier articles, but the difference was small. On average, they found peer review of their COVID-19 article to be of similar quality to peer review of earlier articles.

Discussion and Conclusion

The goal of our research was to better understand the experiences of authors conducting COVID-19 research with regard to preprinting their work, publishing it in a journal, and receiving feedback on their work, both from readers of their preprint and from editors and reviewers of peer-reviewed journals.

Preprint posting

We analysed the results of a survey disseminated among corresponding authors of COVID-19 preprints posted on arXiv, bioRxiv, medRxiv and ChemRxiv in 2020. Two thirds of our survey participants reported that they posted their first preprint during the pandemic. Notably, more than 80% of the surveyed authors expressed the intention to continue preprinting in the future, at least for some of their work. This result may suggest that the pandemic has been a major contributor to structural changes in the scholarly publishing process, indicating that the rising uptake of preprints in this period may not be a temporary trend but may have a lasting impact in the post-pandemic period. Of course, preprinting is not new, and so the impact of the pandemic needs to be set in the context of longer-term increases in adoption of open science practices more generally.

The main factors that motivated respondents to post their COVID-19 articles on preprint servers were the opportunity to rapidly (86.2%) and openly (63.9%) disseminate their research and receive feedback (31.2%). This result aligns well with previous studies. Fraser, Mayr & Peters (2021) conducted a survey of authors of bioRxiv preprints and found that for more than 85% of survey participants the main motivations for preprinting their work were to share their findings more quickly and to increase awareness of their research. In a survey conducted by ASAPbio (2020), more than two thirds of 500 surveyed preprint authors and non-authors mentioned “increasing the speed of research communication” and “preprints are free to read” as the main benefits of preprints. In a study conducted before the pandemic by Chiarelli et al. (2019), 38 key stakeholders including research funding representatives, research-conducting organisations, preprint services, other related service providers, and researchers were interviewed using a semi-structured approach. The study found that the two most significant perceived benefits of preprints, reported by more than 50% of interviewees, were the early and rapid dissemination of research findings and increased opportunities for feedback.

Fraser, Mayr & Peters (2021) also found that for early-career researchers receiving feedback was a more important motivation to preprint their research than for late-career researchers. The same trend can be observed among researchers who participated in our survey: 35.7% of those with fewer than five years of research experience responded that receiving feedback was their primary motivation for preprinting their research, and this proportion decreased with the level of experience of researchers.

Preprint feedback and journal peer review

Preprints have been useful for prompting feedback on COVID-19 research: more than half of the survey respondents reported that they received feedback on their preprint. 41.6% of the feedback was given in an “open” way, e.g., on a preprint server, peer review platform or social media.

Approximately 17% of all authors (n = 673) reported receiving feedback on their preprints directly on a preprint server. Similar to our findings, Fraser et al. (2021) conducted a study on bioRxiv and medRxiv preprints posted during the initial 10 months of the pandemic and reported a comparable rate of feedback and engagement. Specifically, their study revealed that around 16% of these preprints had received at least one comment on a preprint server. Carneiro et al. (2022) reported a lower rate of feedback for preprints posted in 2020 on bioRxiv and medRxiv. During the 7.5 months following the posting date, their study found that only 7% of preprints had received at least one comment on a preprint server.

In our survey, authors who posted their COVID-19 preprints on bioRxiv and medRxiv reported that respectively 50.0% and 38.5% of the feedback on their work was received directly on the preprint server. arXiv and ChemRxiv do not offer the possibility of receiving feedback directly on the preprint server. The results for bioRxiv and medRxiv may suggest significant value in offering a commenting option directly on the preprint server to stimulate open feedback on preprints. Preprint servers that do not have this function might usefully consider adding it. However, considering the challenges mentioned by Ginsparg (2016) regarding “human labor to moderate the comments” and a preference for “drama-free minimalist dissemination” of preprints, preprint servers might also encourage the development and use of other open channels for feedback on preprints (e.g., platforms for preprint peer review).

At the same time, the majority of the feedback reported by our survey participants was received in a “closed” way–privately to the authors. The alignment between our findings and previous studies regarding various aspects of preprint commenting suggests that the high rate of privately provided feedback on preprints observed in our study may not be unusual. This indicates the need for future studies to explore in more detail the extent to which preprints receive private feedback and the motivations for giving feedback privately rather than publicly.

To evaluate the potential of preprint peer review, we asked the surveyed authors about the nature of the feedback given on their preprint. The nature of the comments on bioRxiv was earlier analysed for non-COVID-19 research (Malički et al., 2021), where only 12% of the comments were classified as “full review reports traditionally found during journal review” by a group of independent coders. Another study by Sawyer et al. (2022) analysed about Disqus 500 comments left on the first 1000 preprints indexed in PubMed Central and posted on bioRxiv or medRxiv. This study found that only 11% of comments left on preprints had aspects of a formal peer review process. In line with these earlier findings, our respondents also reported that the majority of the feedback received on preprints did not include detailed comments on the research presented. Of the respondents that received feedback on their preprint, 25.4% indicated that the feedback was of a detailed nature.

According to our respondents, the results section of their paper was altered in major ways in 1.9% of cases as a result of preprint feedback and in 10.1% of cases as a result of journal peer review. In the discussion/conclusion section of their paper, major changes were made in 2.8% of cases as a result of preprint feedback and in 11.3% of cases as a result of journal peer review. This seems to suggest a greater added value of journal peer review compared to feedback on preprints. It also may indicate that making changes as a result of peer reviewer comments when submitting to a journal is seen as less ‘optional’ than making changes in response to feedback on a preprint.

Outlook

Although preprints still constitute only a relatively small proportion of the total publications in many fields, the results of our study show the increasing interest in preprinting among survey participants. Previous studies (Abdill & Blekhman, 2019; Chiarelli et al., 2019; Chung, 2020; Delfanti, 2016; Polka, 2017; Puebla, Polka & Rieger, 2022; Smart, 2022; Vale, 2015) confirm that preprints have already achieved an important position in the scholarly publishing system. A key challenge for stakeholders in the scholarly communication system now is to ensure that preprints are utilised to their full potential.

One aspect of this is preprint peer review (Polka et al., 2022), for which a variety of approaches are emerging. However, they are still in an early stage of development. A workshop on Recognizing Preprint Peer Review brought together various stakeholders from the scholarly communication system (Avissar-Whiting et al., 2023). This resulted in the formulation of a call to action, urging stakeholders to embrace preprint reviewing.

Our study confirms the potential of feedback on preprints to provide valuable comments on research leading to greater efficiency in scholarly publishing. There are various future scenarios in which preprint servers, supported by different reviewing tools and services, could complement journal peer review, or even serve as an alternative to traditional peer-reviewed journals. There is a need for further research to establish robust evidence of approaches that can improve the efficiency and effectiveness of scholarly publishing and can guide future developments.

Limitations

Our study was restricted to four preprint servers, arXiv, bioRxiv, medRxiv, and ChemRxiv, representing around 55% of all COVID-19 preprints posted in 2020 (Waltman et al., 2021). The remaining 45% of the COVID-19 preprints were not considered. COVID-19 research in the social sciences was probably underrepresented. Moreover, the focus of our survey was on authors of COVID-19 preprints. The survey did not include a control group of researchers who did not preprint their COVID-19 research.

Like most survey studies, our research has a self-selection bias, since corresponding authors of COVID-19 preprints who received an email invitation decided themselves whether or not to take part in the survey. We were unable to test the representativeness of the respondents for factors such as the respondents’ country of residence, type of research institution, research experience, and gender in relation to the general population’s distribution, as we lacked data on these variables.

Finally, we did not conduct any further analysis to demonstrate the robustness of the results obtained from questions containing the terms “minor changes” and “major changes”. Rather than explicitly defining the terms “minor” and “major”, we relied on the perceptions and interpretations of participants, as these terms are commonly used in journal peer review processes.

We thank the partners in the COVID-19 Rapid Review Initiative for the pleasant collaboration. We are grateful to Giovanni Colavizza, João Eurico da Fonseca, Serge Horbach, John Inglis, Jessica Polka, Ruy Ribeiro, and Nicolas Robinson-Garcia for testing the survey. We also thank three reviewers for their helpful comments.

Additional Information and Declarations

Competing Interests

Author Contributions

Human Ethics

Data Availability

The authors declare there are no competing interests.

Narmin Rzayeva conceived and designed the experiments, performed the experiments, analyzed the data, prepared figures and/or tables, authored or reviewed drafts of the article, data curation, Validation, Project administration, and approved the final draft.

Susana Oliveira Henriques conceived and designed the experiments, analyzed the data, authored or reviewed drafts of the article, validation, and approved the final draft.

Stephen Pinfield conceived and designed the experiments, authored or reviewed drafts of the article, supervision, Methodology, Validation, Project administration, Funding acquisition, and approved the final draft.

Ludo Waltman conceived and designed the experiments, authored or reviewed drafts of the article, supervision, Methodology, Validation, Project administration, Funding acquisition, and approved the final draft.

The following information was supplied relating to ethical approvals (i.e., approving body and any reference numbers):

Ethical approval to carry out the survey was granted by the Ethics Review Committee of the Social Sciences at the Faculty of Social and Behavioural Sciences of Leiden University.

The following information was supplied regarding data availability:

The data that support the findings of this study are openly available in figshare, except for the free-text responses, which may contain sensitive information.

Rzayeva, Narmin; Oliveira Henriques, Susana; Pinfield, Stephen; Waltman, Ludo (2022). The experiences of COVID-19 preprint authors: A survey of researchers about publishing and receiving feedback on their work during the pandemic. Supplementary data. figshare. Dataset. https://doi.org/10.6084/m9.figshare.21076834.v3.

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
