# Peer review of "The experiences of COVID-19 preprint authors: a survey of researchers about publishing and receiving feedback on their work during the pandemic"

_PeerJ, doi:10.7717/peerj.15864_

## Round 0.1 · original submission · Major Revisions

All the reviewers liked the paper. Two reviewers requested minor revisions and one major revisions. The latter thinks that your conclusions are not adequately supported by your results and you should address this issue in particular in your revision.

·

Basic reporting

Insufficient literature review, and there was a mix of results, methods and discussion in results section that I recommend be separated in appropriate sections. Too many Figures that are not necessary, a tale format wold be more appropriate for this data for univariate analyses.

Experimental design

Insufficient data analysis and exploration. See my detailed comments under 4.

Validity of the findings

Conclusions often go beyond the study design. See my detailed comments under 4.

Additional comments

Dear Authors and Editor,
Thank you for the opportunity to review this manuscript. I enjoyed reading it, and I would like to offer my suggestions for its improvement:
Statements:
1. Authors should list information on: ethics approval, data availability, COI, authors contribution, reporting guidelines adherence, study (protocol) registration, sample size calculation and presentation of the (preliminary) research at conferences.
Title
1. I would suggest specifying surveyed preprints in the title

Abstract
1. Please clearly specify and number the different objectives in the abstract, and report results in the order of the listed objectives.
2. Your first result sentence in the abstract (observed a high rate of new adopters) is not captured in any objectives listed. This should be amended.
3. In the abstract you state we observed, and saw, yet the study was a survey – therefore it would be more appropriate to state respondents reported…
4. The final sentence of conclusions seems out of place as you reported only a quarter of preprints received feedback resembling peer review. How did you classify this feedback as peer review, as your survey questions did not equate any feedback answer options as resembling peer review. Furthermore, comparing feedback with peer review seems comparing different things, and without good rationale I find this comparison incorrect. You study was unable to assess if feedback received covered the same feedback as was obtained through peer review, nor did the authors change the paper before peer review based on that feedback, or during the peer review process.
Introduction
1. Please rewrite the introduction to clearly state previous studies on Attitudes, behaviors and experiences of authors on covid and non-covid preprints. Additionally, you state 3 main objectives and none have been covered with an extensive literature review of those 3 objectives.
2. Please make sure that the objectives listed are in alignment with those in the abstract

Methods
1. Please improve the methods section for replicability: Please clearly sate how were articles downloaded, i.e. how were authors and their email information collected – using which code and de-duplication methods, how were they stored, and please share invite and reminder (if any) email texts. Please also specify random number generator used.
2. Please also use CHERRIEs to cover all items for online surveys stated in a reporting guideline. https://www.ncbi.nlm.nih.gov/pmc/articles/PMC1550605/ or another RG if you find it more appropriate.
3. Pease comment on how was the collection of emails approved by the ethics board, and where were the data with names and emails stored - as this information falls under GDPR rules as you did not share that database on figshare.
4. You stated: These respondents seem to have misunderstood the question – Please specify why this question had more options than the 4 servers you collected emails from. Additionally, there have been cases of authors who have posted their preprints on multiple servers, so this interpretation is only one of many. Please also specify did you and why include or not include rest of their answers in the final results, as if as you say authors misunderstood this question, they might have misunderstood many others.
5. As you know the total population, and you could have estimated the gender based on author names – please specify why you did not use weights to adjust for survey results. Generalizability of your results is questionable and needs to be commented on. What about non-respondents and their opinions. Standard survey biases are not discussed and this needs to be addressed. Did you conduct wave analysis to explore potential responder biases.
6. Please also specify if any correlation or regression analysis were done to determine differences based on characteristics of responses and their answers, and if not why not.
7. Please list details on how was qualitative data analyzed and by whom.
8. Finally, please also list how many respondents did not answer all of the survey questions, i.e. and are results reported per number of respondents of individual questions or were some answers imputed.

Results
1. I personally find the number of figures and tables too big for this article, and the way data was presented to not follow standards of my field (e.g. Basic Statistical Reporting for Articles Published in Biomedical Journals: The “Statistical Analyses and Methods in the Published Literature” or The SAMPL Guidelines”). Please always report n and %, do not use pie charts and bars for data that is best reported in tables as n and %, or specify in methods which standards of reporting are followed. In same cases you reported SD and in others not, was the data normally distributed for this – this should be explained under stat. analyses? If you instead used +- signs for CI, this should be stated.
2. As you shared the full list of questions, I would recommend that in the supplementary file you report results for the questions in the order they are in the survey, with n and % for all options. This would allow for an easier follow of results.
3. Regarding the question – when did you first post it – I believe that the fact that some of these servers did not exist before 2017 needs to be taken into account. Additionally, you analyzed this question only based on server where they posted, and not their age. Regression to see differences between field vs age or other factors are needed.
4. You stated: It was a priority for us to understand the authors’ motivations – I recommend all motivation and interpretations like this one be removed from results, and reported in methods/ discussion.
5. You stated - expressed by many of the other survey participants – please specify n and %, do not use many, most or similar expressions without data.
6. We listed motivations in the survey based on the findings of a previous qualitative study (Chiarelli et al., 2019). – this needs to be stated in methods and likely introduction on previous research on this topic.
7. Comments…by some respondents…As item 5, please report n, and make it clear in methods or results when dissenting opinions were also presented
8. Remarkably – please remove this interpretation. It doesn’t seem appropriate in regards to previous studies on this topic.
9. This suggests that the pandemic has led to long-term changes in authors publication practices – I find this interpretation one of many possible, and would remove it from results. This is an issue that can be mentioned in discussion. It is also possible that open science movement, as well as requirements of institutions or funders played a role in this, as well as the ability to claim primacy of ideas - all that has been reported in previous studies.
10. Since receiving feedback is often argued to be one of the key advantages of posting preprints – This sentence requires citation, and should not be in resuts as items listed previously
11. Other studies besides Kodvanj et al, including my own studies (https://biochemia-medica.com/en/journal/31/2/10.11613/BM.2021.020201/fullArticle, https://www.biorxiv.org/content/10.1101/2022.11.23.517621v1.full, have shown rates of comments for preprints on bioRxiv and Medrxiv, and likely should be included in the literature review. As above, I would recommend however these are discussed in intro and discussions, not results.
12. You stated: “we asked authors to report whether they received feedback that resembled peer review performed by a journal” – please correct this, no question in the survey asked this, instead you interpreted it based on the answer that did not mention similarity to peer review. Based on how the question reads, it could also be interpreted that you defined options 1 to 4 as peer review, and rest not.
13. You stated - due to the limited number of responses per preprint server, statistical uncertainty is high – please specify what stat. method or threshold you used for this, and which other questions/results are affected by high uncertainty. Which methods were used to compare differences between servers, and which co-factors were assessed in these analyses?
14. Have you explored correlation between making changes and papers being rejected, or other questions in the survey that might indicate quality of articles
15. Finally, as survey had open ended options, please report how many respondents you had per each such questions, and how were they analysed/coded and how many different categories you found for those answers.
Discussion
1. Our survey shows – please use past tense, as the study is completed. I would also recommend that discussion starts with the main summary of results and comparison of those results with previous studies on the topic. It is not clear what new knowledge is gained from this study.
2. As mentioned above, your study was not designed to explore the reasons why authors plan to preinrt in the future, and the sentence - pandemic has caused structural changes in the scholarly publishing – is just one of many possible answers, other options should be listed, as well as statement that this is not an answer assessed by the survey
3. In our research, we also see – please be consistent in reporting, this was a survey, and respondents did or did nor report something.
4. representing around 55% of all COVID-19 preprints posted during 2020 – this requires citation, where does this number come from, there are more than 60 preprint servers out there.
5. Would recommend having a dedicated limitations subtitle, and expand the discussion and limitations greatly to comment on generalizability of results, as well as correlation/regression analyses, influence of gender of respondents, and differences between countries, age etc. Survey results seem to be insufficiently explored in regards to data that was collected.
6. Finally, many interpretations stated in the paper are listed with the tone and phrases that are not adequate for study in question. This was a survey, and while you as authors may think that these results show or suggest, I would recommend these terms are amended to state might show, or may suggest as many alternative options exist and this study was not aimed to answer those questions. More emphasis should be put on what respondents stated, rater than on how the authors interpret what this means – when those questions were not asked of authors.


In hopes may comments can help you improve your manuscript,
Kind regards,
Mario Malicki

Reviewer 2 ·

Basic reporting

In this manuscript, the authors set out to assess how preprints have been used by scientists during the pandemic from an author perspective.
The paper is particularly timely and relevant and the topic is rightfully put into context by the authors at the beginning of their manuscript. It was overall an easy and very interesting read and I quite enjoyed both reading and reviewing this manuscript.

I will list below what I believe are points on which the authors can improve their manuscripts and I hope that they address those comments.

The type of visual representation for figure 12 & 13 makes the data slightly difficult to read. I would recommend that the authors use this kind of visual representation instead [A]. Similarly the color-coding used on Fig. 1 does not allow readers to read percentages for "other". I would also recommend generating and integrating PDFs of images instead of PNGs/JPGs (not having submitted myself to the journal I am not sure that the authors have control over the format though), because the images appear very blurry. For each figure, the authors should mention what the error bars represent (having it in the caption is easier).

A main limitation of the research, I would argue, is that the authors did not define the terms "major changes" & "minor changes". While one can hope that the definition used has been consistant across the questions about preprint feedback and journal feedback, it might make the results sometimes quite difficult to interpret. Indeed, I would personally not consider rewriting to ever be a major change, even the full paper needs to be re-structured, but some people might argue differently. If they do, it could be that journal reviewers asked for restructuring in a review which the authors would then consider major but such comments are unlikely to be made for preprints for which structure and format are likely to matter less. This is not sufficiently detailed in the manuscript and might warrant some additional analysis to show that the results obtained are robust across different characterization. I am however not sure it is possible to do with the data that the authors have gathered. I could be convinced also by a better and more emphasized explanation of this limitation in the manuscript and its impact on the results. I leave this comment up to the editor and other reviewers to discuss on how critical this could be and how it must be addressed.


"In our research, we also see that most of the feedback did not contain elements characteristic of journal peer review." This indeeds reflects past research on the topic, although I would have liked the authors to explain or give hypothesis on why this might be the case. There are several factors that could explain the lack of "elements characteristic of journal peer review". The first is the obvious fact that preprint reviews do not have to formally review the entirety of a paper and could instead focus on specific elements. In doing so, a review would therefore not appear as complete as traditional reviews from journals, but that does not particularly assess the quality of the comment/review made. Another factor is the fact that the platforms do not necesseraly entice such long comments, I would argue, when compared to a traditional peer review invitation. Another aspect is that comments made by readers cannot be valued in academic CVs as much (at least traditionally) and this therefore does not foster preprint comments to be as complete as journal reviews.

I would argue that perhaps other aspects of preprints should be discussed in the manuscript, at least to open the discussion points or findings from the authors. The first one should be to discuss the credibility of preprints and past research on the topic [B], and the second aspect would be to discuss the potential misuse of preprints that has been made during the pandemic [C]. Discussing these two points, even just as "openers/pointers" would give future readers some insights on other research that have been conducted around preprint and in particular for [C] around usage of preprints as potential tools by the public or journalists.

Overall, I am quite positive about the submission and I think it is an interesting paper that, after some revisions, should be published. I look forward to reading the published version. Having never submitted nor reviewed for this journal before, I would be more than happy to discuss my recommendation with the editorial team with respect to their expectations for the journal.

References

[A] https://r-graph-gallery.com/202-barplot-for-likert-type-items.html
[B] https://royalsocietypublishing.org/doi/full/10.1098/rsos.201520
[C] https://bmcmedresmethodol.biomedcentral.com/articles/10.1186/s12874-021-01304-y

Experimental design

See the review above, the main point of criticism fits both this box and the next one.

Validity of the findings

See the review above, the main point of criticism fits both this box and the previous one.

Additional comments

See the review above

·

Basic reporting

This is an interesting study analysing the experience of authors of Covid-19 preprints in 2020, with particular focus on feedback on preprints versus journal articles - an area that needs to be further explored - and a comparison of peer review conducted before and during the pandemic. I greatly appreciate the authors’ work.

As outlined in the abstract and the following survey design, the survey relates specifically to feedback on Covid-19 publications and the comparison of preprints and journal articles. I would suggest highlighting this in the title, which is quite general in its current wording.

The questionnaire and the survey data is available in figshare.

Experimental design

The survey seems to me well applied. Everything is clearly laid out.

Two things I'd like to ask, because I'm curious about this:

Among the characteristics of the respondents (shown in Figure 1), have you checked whether there are any striking correlations, e.g., in terms of choice of a respository, motivation for preprinting, or experience with feedback and subsequent changes in results?

Is there a systematic evaluation of the free-text responses in terms of qualitative coding that can be presented for in-depth analysis or added on figshare?

Validity of the findings

I would like to add some recommendations on the Results and Discussion sections:

I wonder why respondents could select to that they post a paper on a preprint server for the first time during 2020 and 2021 (line 196 and question 8 in the questionnaire), because as described earlier, corresponding authors of preprints posted in 2020 were invited to participate in the survey.

I suggest moving the reference to the other platforms (lines 199-203) to a footnote or even deleting it. This detail has already been mentioned (lines 157-160). Instead, some references to how established the other four preprint serves are (e.g., the long tradition of arXiv compared to the others) could be included.

In line 316, for the sake of completeness, it would be worth adding that nevertheless, overall, not that much has been changed in the preprints.

In the case of Figure 10 (description in lines 410-415), I cannot exactly understand the methodological procedure respectively the advantage in representation compared to Figure 7 and 9. At least I suggest to explain this in more detail and adjust the titles of Figures 7, 9 and 10 to better understand what proportion of what is meant.

In the description of Figure 11 (lines 423-425), major changes were also reported for “changes to the structure”, with a slightly higher percentage for “inclusion of results from additional data collection”.
Perhaps the heading “Limitations” should be added to line 543 for clarity.

If there are any other issues specifically related to the survey results that should be explored in the future, they could be added following line 542. In this context, as a reader, I would also be interested to know if there are other subsequent parts of the COVID-19 Rapid Review Initiative, since this survey is mentioned as an outcome of those evaluations, as described above (lines 86-96).

Additional comments

In one reference (line 627), some authors are named twice.

---

## Round 0.2 · Minor Revisions

The reviewer that required major revisions is now much happier with your paper and has a few minor comments for you to address.

·

Basic reporting

Improved since version 1

Experimental design

Improved since version 1, limitations listed.

Validity of the findings

Improved since version 1, limitations listed. Would still prefer that regression analyses were conducted,

Additional comments

Thank you for the detailed rebuttal letter, and making extensive changes to the manuscript. I find it a much better read now. The following are the final remarks you might take into consideration.

I would recommend removing or expanding the following sentence - This feedback rate is much higher than in other studies and paragraphs following it. (You stated: Of the respondents (n=673), 52.8% (±3.8%) reported receiving feedback on their preprint.). Yours seems to be the only study that directly surveyed this of authors. And so please emphasize that, and state more clearly that the 19% received feedback on social media, is directly comparable to the Kodvanj study which analysed almetrics, and similar to 16% of the Fraser, N. et al. (2021) study ‘The evolving role of preprints in the dissemination of COVID-19 research and their impact on the science communication landscape’. And please compare commenting on servers with Chaiara study or other appropriate studies. In other words, in the discussion emphasize that the specific feedback rates reported in your survey are similar to studies that analysed altmetrics or comments on servers, but that yours may be the first to report on the rates of closed feedback which was found to be X%, (if this is not the case compare it to those that did also survey the authors and asked about feedback).
I am likely biased, but I find your results of 50% of feedback received as closed to be the most exciting finding of your survey. And that is why I would like you to expand upon this more in results and discussion. For example, if 50% received feedback, and 50% of that feedback was closed, would you say that your data indicates 25% of preprints get closed feedback. And how big were the differences between the servers? Would the worse case scenario be that as you only had 7% of response rate, that in the extreme where all who did not receive feedback, also did not answer your survey, it could be than that only 3.5% of 12000 received feedback, of which 1.7% closed one? Should both interpretations be presented as we can not be sure how representative are your respondents to the 12000 you sent email to? I would hope that as the rates of social media or direct comment engagement are the same as in those other studies, that the 25% closed engagement is representative, and that it should be emphasized more. And you might recommend that further survey studies on different (and the same) servers explore this question or monitor it over time to determine if they are lower for non/Covid studies, as has been seen in commenting on medRxiv.

---

## Round 0.3 · accepted · Accept

Thank you for addressing the comments of the reviewer. I am now happy to accept your paper for publication.